# Drug dosage modifications in 24 million in-patient prescriptions covering eight years: A Danish population-wide study of polypharmacy

Cristina Leal Rodríguez[1], Amalie Dahl Haue[1,2], Gianluca Mazzoni[1], Robert Eriksson[1,3], Jorge Hernansanz Biel[1], Lisa Cantwell[1], David Westergaard[1,4], Kirstine G. Belling[1], Søren Brunak[1]*

**1** Novo Nordisk Foundation Center for Protein Research, Faculty of Health and Medical Sciences, University of Copenhagen, DK-2200 Copenhagen, Denmark, **2** The Heart Center, Rigshospitalet, Copenhagen University Hospital, DK-2100 Copenhagen, Denmark, **3** Department of Pulmonary and Infectious Diseases, Nordsjællands Hospital, DK-3400 Hillerød, Denmark, **4** Department of Obstetrics and Gynaecology, Copenhagen University Hospital Hvidovre, DK-2650 Hvidovre, Denmark

* soren.brunak@cpr.ku.dk

**Data Availability Statement:** Following ethical approval data can be made available for use in secure, dedicated environments via application to

## Abstract

Polypharmacy has generally been assessed by raw counts of different drugs administered concomitantly to the same patients; not with respect to the likelihood of dosage-adjustments. To address this aspect of polypharmacy, the objective of the present study was to identify co-medications associated with more frequent dosage adjustments. The data foundation was electronic health records from 3.2 million inpatient admissions at Danish hospitals (2008–2016). The likelihood of dosage-adjustments when two drugs were administered concomitantly were computed using Bayesian logistic regressions. We identified 3,993 co-medication pairs that associate significantly with dosage changes when administered together. Of these pairs, 2,412 (60%) did associate with readmission, mortality or longer stays, while 308 (8%) associated with reduced kidney function. In comparison to co-medications pairs that were previously classified as drug-drug interactions, pairs not classified as drug-drug interactions had higher odds ratios of dosage modifications than drug pairs with an established interaction. Drug pairs not corresponding to known drug-drug interactions while still being associated significantly with dosage changes were prescribed to fewer patients and mentioned more rarely together in the literature. We hypothesize that some of these pairs could be associated with yet to be discovered interactions as they may be harder to identify in smaller-scale studies.

## Author summary

Polypharmacy continues to grow in importance because of multimorbid, aging populations. In healthcare, polypharmacy is well-known for its detrimental consequences related to adverse events. Polypharmacy is also a key challenge included in the precision medicine agenda. So-called precision dosing is important for addressing concomitant diseases and

the Danish Regions and the Danish Health Data Authority (https://www.enindgangtilsundhedsdata.dk/en/services/ansoegningsportalen). To access the data, first obtain research credentialing requirements (https://sundhedsdatastyrelsen.dk/da/english/health_data_and_registers/research_services). This study was approved by the Danish Patient Safety Authority (3-3013-1731 and 3–3013–1723), the Danish Data Protection Agency (DT SUND 2016–48, 2016–50, 2017–57 and UCPH 514-0255/18-3000), and the Danish Health Data Authority (FSEID 00003092, FSEID 00003724, FSEID 00004758 and FSEID 00005191). The Danish Health Data Authority can be contacted at kontakt@sundhedsdata.dk. Stan (v2.21) (72), Python (v3.5.4), and R (v.3.4.0) were used for data processing and statistical analysis and the code is available upon request to the Big Data Management Platform at the NNF Center for Protein Research (contact@cpr.ku.dk). The provided supplementary data are non-person-sensitive summary level data.

**Funding:** This work was supported by the Novo Nordisk Foundation (grants NNF17OC0027594, NNF14CC0001 and NNF18SA0034956) and the Danish Innovation Fund (grant 5153-00002B). These foundations contributed to the financing of salaries for all the authors. The funding bodies had no role in the design and conduct of the study; collection, management, analysis, and interpretation of the data; preparation, review, or approval of the manuscript; or the decision to submit the manuscript for publication.

**Competing interests:** I have read the journal's policy and the authors of this manuscript have the following competing interests: S.B. reports ownerships in Intomics A/S, Hoba Therapeutics Aps, Novo Nordisk A/S, Lundbeck A/S and managing board membership in Intomics A/S outside the submitted work. The other authors have declared that no competing interests exist.

medications, and their impact on treatment responses. In this work, we present a study aiming at investigating this problem using real-world data from ~185 million treatment episodes leading to significant statistical power across drug-cocktails. We introduce a comprehensive analysis of dosage changes aiming to identify those significant pairs that may influence each other. More than half of the identified drug pairs were associated with readmission, mortality or longer stays and we also observed major differences in relation to disease and laboratory tests. Under the premise that drug-drug interactions are manageable through patient monitoring and dosage adjustments, we collected and cross-referenced information from a wide range of clinical and bioinformatics drug-drug interaction databases that could be related to pairs associated with dosage changes. Overall, this work shows how distinct, changing dosage patterns can facilitate the identification of drug-drug interactions in the context of polypharmacy complementing longitudinal analyses of disease progression.

## Introduction

Polypharmacy is a widely observed phenomenon; especially among hospitalized patients, where it has been associated with in-hospital mortality and readmission [1]. These associations are partly explained by the fact that polypharmacy increases the risk of compromising the drug response through drug-drug interactions (DDIs) [2–7]. DDIs may influence the efficacy or the risk of toxicity of a drug leading to an increased risk of adverse drug reactions (ADRs) [8,9]. Although drugs are being tested repeatedly in clinical trials, pre- and post-marketing, knowledge of dosage-adjustments when drugs are administered concomitantly is limited [10].

Large-scale clinical trials of polypharmacy are difficult to carry out and it has not previously been characterized in a population-wide setting with respect to dosage-adjustments, which is of obvious importance to expected DDIs and ADRs. Instead, polypharmacy has typically been addressed in the elderly or in selected phenotypes with reference to surrogate measures like the medication burden index [11–13] and the comorbidity-polypharmacy score [14,15], which are useful, but do ignore many aspects of drug treatment contexts [11]. Another common approach is to assess polypharmacy by means of standard evaluation of potential pharmacokinetic inhibitors and inductors in relation to a known DDI. In this case, only major CYP450s and drug transporters are tested based on likely interacting drugs that are known to be prescribed concomitantly in selected target populations [16–18]. Thus, new methods are needed for characterizing polypharmacy in a data-driven manner.

Studies on polypharmacy, including identification of dose-dependent ADRs, have been conducted for entire populations and for selected patient groups using electronic health records (EHRs) [19–27]. However, their focus has primarily been inappropriate prescriptions, improvement of patient adherence, reduction of readmissions, ADR detection, and examination of the potential for deprescription [28–30]. Thus, evidence for the relationship between polypharmacy and dosage-adjustments is sparse [22]. Pharmacogenomics is an emerging and cutting-edge field aiming to understand drug response variability [31], but in order to systematically study their associations with genetics, it is essential first to map the polypharmacy-related dosage changes observed in real world settings including their associations to outcomes [32]. These aspects further highlight the need for studies that quantify polypharmacy with respect to drug-dosage adjustments.

In response, we applied Bayesian inference to characterize the complexity of polypharmacy and identify drug pairs subject to more frequent dosage adjustments in a population-wide

setting. We describe their association to outcomes, diagnoses, blood tests, and classify them based on whether they have been labelled as DDIs previously and share pharmacokinetic properties. Based on the results, we argue that the application of Bayesian inference in the context of EHRs reveals unknown aspects of polypharmacy that are notoriously difficult to recognize in smaller, highly selected, populations using more traditional methods.

## Results

### In-hospital drug use and polypharmacy

In this observational study, we used in-patient prescription and admission data from Eastern Denmark (50% of the Danish population) containing data from all patients hospitalized in an eight-year period (1,069,873 patients, years 2008–2016), see **Table 1**. Among the 24,379,285 inpatient prescriptions, 902 distinct drugs were prescribed to at least 50 patients; and these drugs were concomitantly administered with up to 857 other drugs (**Fig 1**). The therapeutic groups that were prescribed to most patients were analgesics (N02), antibiotics (J01), and antithrombotic agents (B01). These were also the ones with the highest number of concomitant medications (Pearson $\rho$: 0.93, 95% confidence interval CI: 0.91–0.96, p-value = $1.70 \times 10^{-37}$). The median number of uniquely prescribed drugs per patient was six. The degree of polypharmacy varied across different admission reasons (primary diagnoses) with a median of drugs ranging from three to eight (**S1 Table**). Polypharmacy was prevalent in all age groups and the number of different drugs correlated positively with age ($\rho$: 0.40, 95% CI:0.39–0.40, p-value $< 2.2 \times 10^{-16}$, **S1 Fig**).

### Co-medication pairs and likelihood of dosage adjustments

Of the 857 drugs that were administered concomitantly, 413 fulfilled the selection criteria for using the Bayesian inference model (**S2 Fig** and **Methods**). The 413 drugs combined into

**Table 1. Hospital admission characteristics (n = 3,161,647).**

| | |
|---|---|
| **Number of patients** | 1,069,873 |
| **Admissions/patient, median [IQR]** | 5 [2–10] |
| **Age, median [IQR]** | 59 [36–73] |
| Women (%) | 1,702,030 (54) |
| **Number of admissions with status death (%)** | 63,609 (2.0) |
| **Length of stay (days), median [IQR]** | 2 [1–5] |
| **Number of drug prescriptions** | 24,379,285 |
| **Number of drug prescriptions/admission, median [IQR]** | 8 [3–14] |
| **Number of unique prescribed drugs, median [IQR]** | 6 [3–11] |
| **Most frequent drug classes, number patients (%)** | |
| Analgesics, anilides (N02BE) | 774,305 (72) |
| Analgesics, natural opium alkaloids (N02AA) | 462,535 (43) |
| Anti-inflammatory/antirheumatic, propionic acid derivatives (M01AE) | 371,309 (35) |
| Antithrombotic agents, heparin group (B01AB) | 352,758 (33) |
| Antibacterials, second-generation cephalosporins (J01DC) | 322,073 (30) |
| Drugs for acid related disorders, proton pump inhibitors (A02BC) | 262,051 (24) |
| Corticosteroids, glucocorticoids (H02AB) | 252,544 (24) |
| Antiemetics, serotonin antagonists (A04AA) | 237,001 (22) |
| Antithrombotic agents, platelet aggregation inhibitors (B01AC) | 224,017 (21) |
| Analgesics, other opioids (N02AX) | 218,377 (20) |

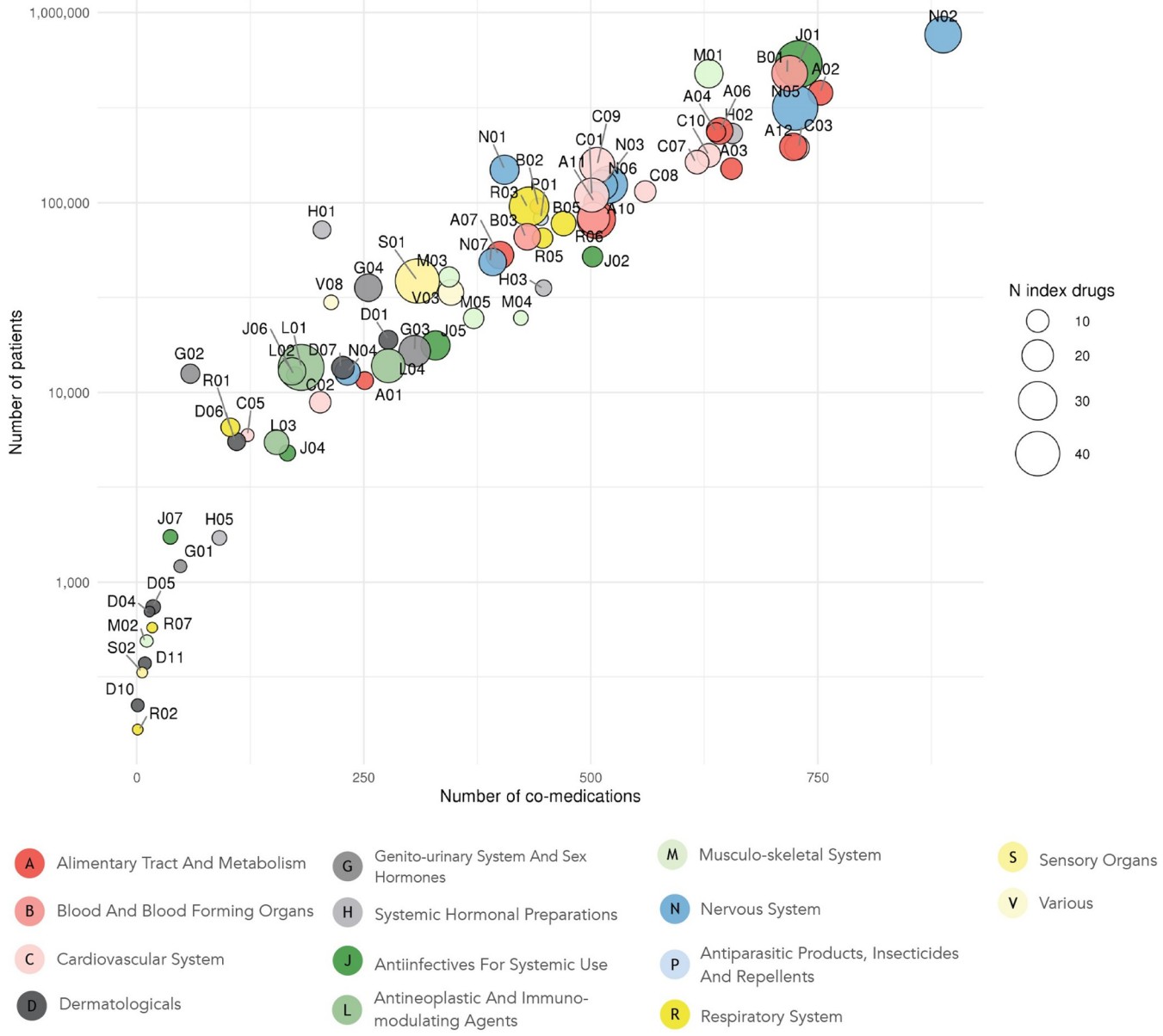

**Fig 1. Prevalence and number of co-medications across therapeutic groups (circles).** Circles are coloured based on the drugs' ATC anatomical group and sized proportional to the number of index drugs within each therapeutic group. Therapeutic groups with one index drug were not included (n = 10). ATC: Anatomical Therapeutic Chemical classification system.

77,494 different co-medication pairs (i.e., an index drug and a co-medication). There were a total of 3,993 dosage-adjusted co-medication pairs (309 distinct index drugs), defined by the subset of co-medications pairs that had a significant association (i.e. OR >1) for dosage adjustment of the index drugs (**S2 Table**).

Next, we characterized the co-medication pairs in relation to the therapeutic group of the index drug, and the anatomical group of the co-medications. While co-medications appeared homogeneously distributed across therapeutic groups (**Fig 2A**, left and **S3 Fig**), dosage-adjusted co-medication pairs were dominated by nervous system (N), cardiovascular system (C), and alimentary tract and metabolism system (A) co-medications (>60%) (**Fig 2A**, right).

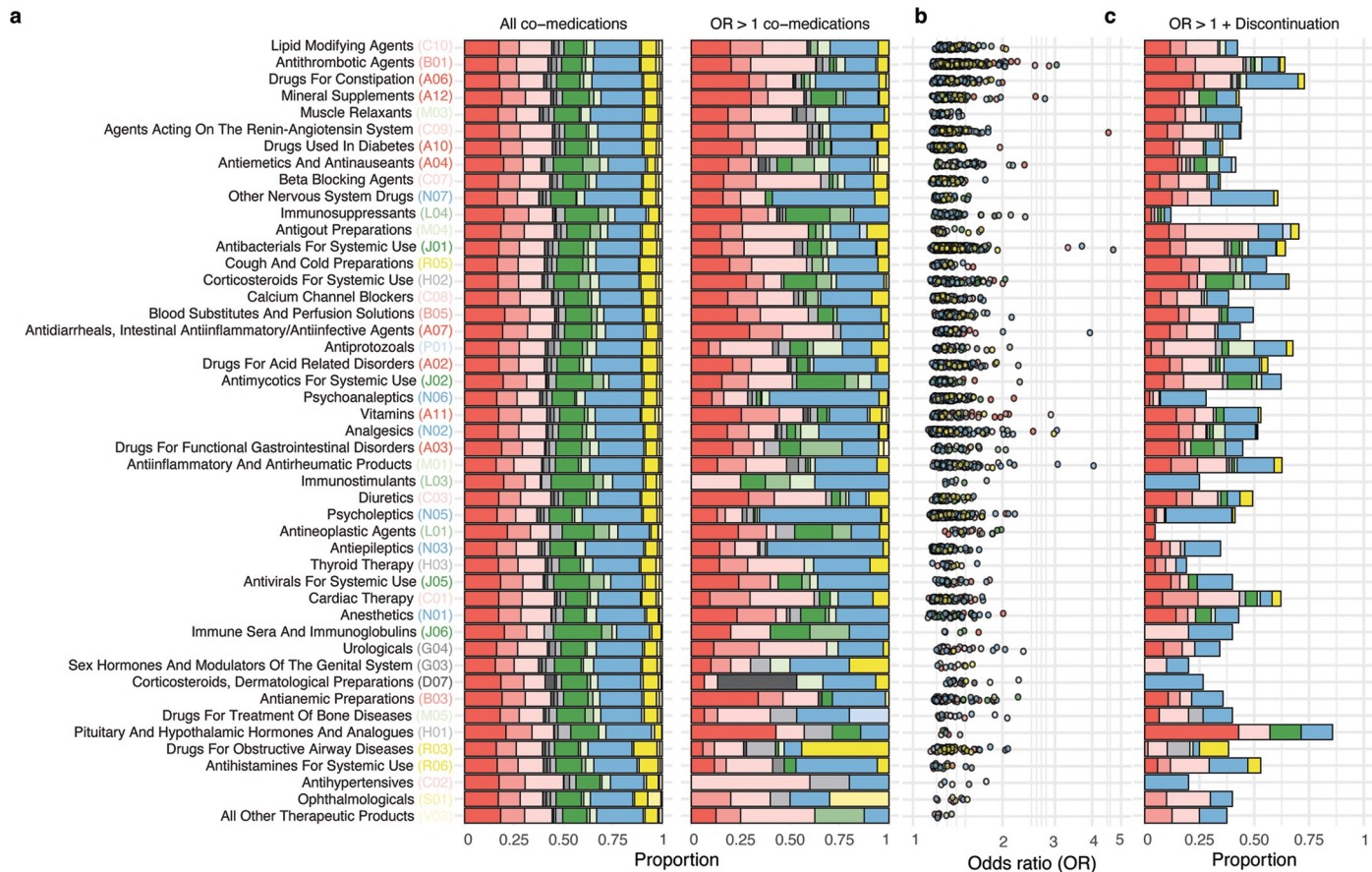

**Fig 2. Representation of co-medications and co-medications associated with dosage adjustments across therapeutic groups. a** Distribution of all co-medication pairs (left) and dosage-adjusted co-medication pairs (with ORs >1, right). Therapeutic groups of index drugs (vertical axis) are ordered by the proportion of co-medications associated with dosage adjustment, lowest in the bottom. **b** OR distribution where each point represents a dosage-adjusted co-medication pair. ORs >3 are observed for index drugs in the therapeutic groups A07, B01, B02, C09, J01, N02 and M01. **c** Proportion of dosage-adjusted co-medication pairs associated with discontinuations (cessation of index drug during concomitant treatment). Vertical axis and colour legend shared for a, b and c. Colour according to therapeutic group of the co-medication (see Fig 1 for legend). Only therapeutic groups represented by more than one index drug and at least five dosage-adjusted co-medication pairs are shown (47 therapeutic groups and 3,944 dosage-adjusted co-medication pairs). OR: odds ratio.

Co-medication pairs where index drugs were classified in the therapeutic groups psycholeptics (N05); psychoanaleptics (N06); antiepileptics (N03); other nervous system drugs (N07); drugs for obstructive airway diseases (R03); corticosteroids, dermatological preparations (D07); and antihypertensives (C02) had co-medications of the same anatomical group in >40% of the dosage-adjusted co-medication pairs. In contrast, for dosage-adjusted co-medication pairs where the index drug was classified as antiemetics and anti-nauseants (A04), the associated co-medication groups were more evenly distributed. Dosages of immunosuppressant (L04) and antimycotic (J02) index drugs were often adjusted when co-medicated with anti-infective drugs (J) (**Fig 2A**, right). Dosage-adjusted co-medication pairs where index drugs were classified as antithrombotic agents (B01), agents acting on the renin-angiotensin system (C09), antibiotics (J01), analgesics (N02), and anti-inflammatory and antirheumatic agents (M01) had the highest ORs (OR >3) (**Fig 2B**, **S4 Fig**).

All index drugs belonging to antithrombotic agents (B01, e.g. dalteparin, enoxaparin, aspirin and clopidogrel), drugs acting on the renin-angiotensin system (C09, e.g. ramipril, losartan, enalapril and tandolapril), and lipid modifying agents (C10, e.g. atorvastatin, simvastatin, rosu-vastatin and ezetimibe) were subject to dosage adjustments with most of their co-medications.

Yet, there was no clear pattern for all therapeutic groups widely co-administered (see **Fig 1**) and the proportion of index drugs likely to be adjusted with co-medications (e.g., N02, A02, J01, C03, N05). Three therapeutic groups had a low co-medication burden (see **Fig 1**) and no co-medications associated with dosage adjustments, i.e., vasoprotectives (C05), antibiotics and chemotherapeutics for dermatological use (D06), and nasal preparations (R01) (**S3 Fig**).

Additionally, 1,949 (49%) dosage-adjusted co-medication pairs were more likely to be stopped during concomitant treatments (**S3 Table**). Dosage-adjusted co-medication pairs where index drugs were classified in the therapeutic groups pituitary and hypothalamic hormones and analogues (H01), drugs for constipation (A06) and antigout preparations (M04) showed the greatest percentage of discontinuation pairs (>70%); especially with co-medications belonging to the anatomical groups alimentary tract (A), nervous (N) and cardiovascular (C) systems, respectively (**Fig 2C**).

## Clinical characterisation of dosage-adjusted co-medication pairs

To assess the clinical meaning of the dosage-adjusted co-medications pairs, we analysed their association with clinical outcomes, diagnoses, and abnormal blood test results (for details, see **Methods**). **Fig 3** presents the number of dosage-adjusted co-medication pairs associated with the outcomes across the different therapeutic groups. Among the 3,993 dosage-adjusted co-medication pairs, 2,412 (60%) associated with either length-of-stay (LOS) (1,243; 31%), mortality (414; 10%) or readmission (263; 7%) (**S4 Table**). Most of the co-medications that associated with LOS belonged to the anatomical group nervous system (N). Index drugs belonging to the therapeutic group antibiotics for systemic use (J01) comprised many dosage-adjusted co-medication pairs that associated with readmission and mortality. For readmission, index drugs classified as antibiotics for systemic use (J01) were dominated by co-medications classified in the anatomical group nervous system (N); whereas co-medications belonging to the anatomical groups alimentary tract and metabolism (A) and cardiovascular system (C) dominated in dosage-adjusted pairs associated with mortality. Similarly, dosage-adjusted co-medication pairs with index drugs belonging to the therapeutic group antithrombotics (B01) were commonly associated with readmission and mortality, although representing a small percentage of the identified dosage adjusted pairs (12% and 7%, respectively, **Fig 3**).

Generally, the diagnoses that correlated with the most dosage-adjusted co-medication pairs were cardiovascular and metabolic diseases, such as primary hypertension, dyslipidaemia, and ischemic heart disease (376 co-medication pairs) (**S5 Fig and Fig 5, S5 Table**). Accordingly, the blood test results that correlated with most co-medication pairs were metabolic markers, markers of coagulation and cardiac biomarkers. For example, the highest number of dosage-adjusted co-medication pairs correlated with troponin T (329 pairs), estimated glomerular filtration rate (eGFR; 308 pairs) and international normalised ratio (INR; 288 pairs) (**S6 Fig, S6 Table**).

To further characterize the most prevalent dosage-adjusted co-medication pairs in relation to clinical characteristics and clinical outcomes, we focused on four therapeutic groups that are generally prescribed to patients with a wide range of morbidities: antidiabetics (A10), antineoplastic drugs (L01), antithrombotic agents (B01), and lipid modifying agents (C10) (**Fig 4**). There was a large overlap among top-ranked diagnoses that associated with drugs used in diabetes (A10), antithrombotic agents (B01) and lipid modifying agents (C10), whereas antineoplastic agents (L01), non-surprisingly, primarily associated with neoplasms (**Fig 4**). In contrast, there was a greater variation among the blood tests that associated with co-medication pairs in the four therapeutic groups. Interestingly, none of the co-medication pairs in the antineoplastic agents group associated with mortality (**Fig 4**).

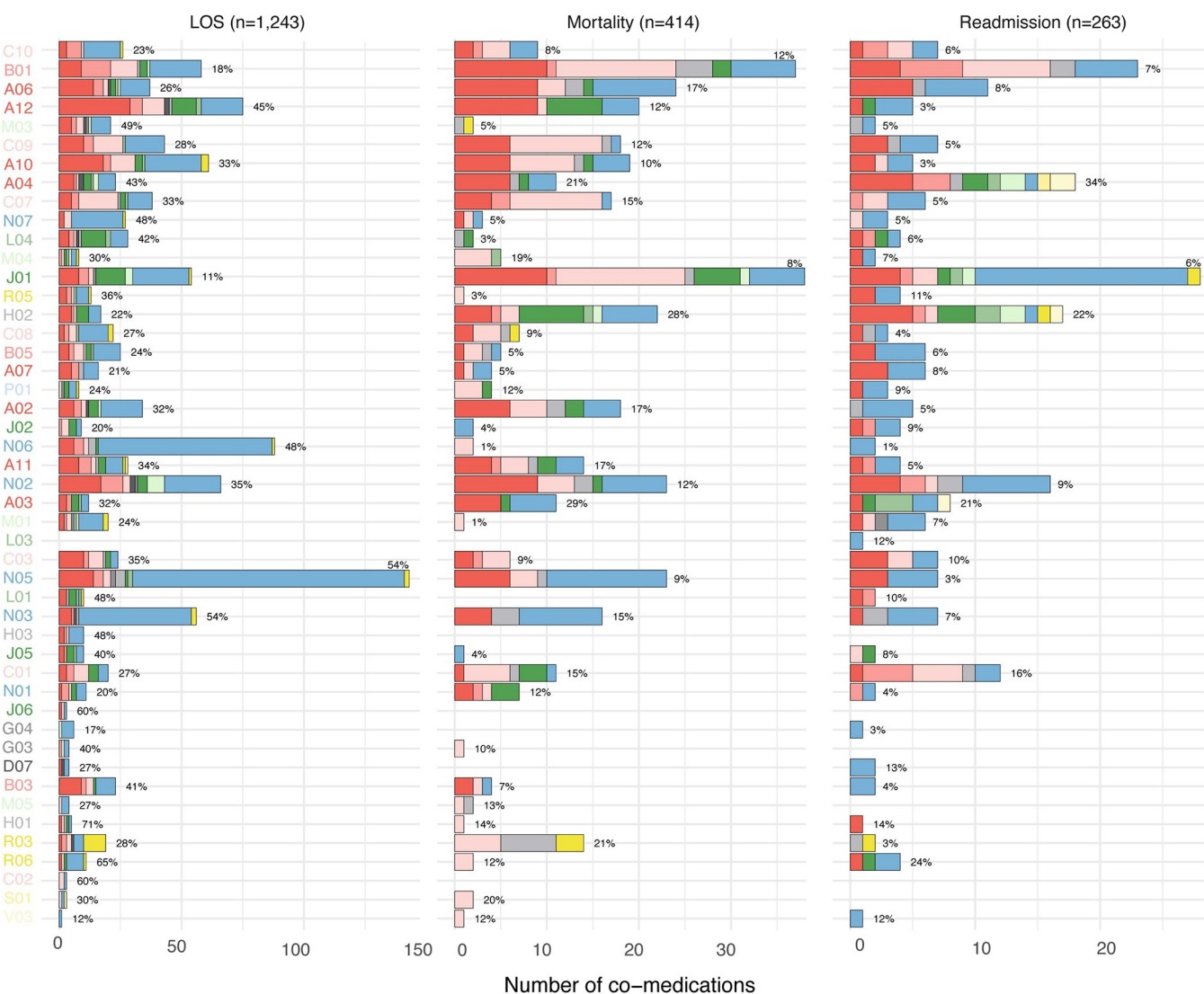

**Fig 3. Outcomes associated with dosage-adjusted co-medication pairs.** Bar plots representing the number of dosage-adjusted co-medication pairs positively associated with 30-days readmission, 30-days all-cause mortality and LOS. Numbers show the percentage of co-medication pairs from the total associated with dosage adjustments for each group. Colour according to therapeutic group of the co-medication. Colour and order of therapeutic groups as in Fig 3 (see **Fig 1** for legend). LOS: Length-of-stay.

## Dosage-adjusted co-medication pairs and evidence of drug-drug interactions

To examine if dosage-adjusted co-medications pairs could be linked to DDIs, the 3,993 dosage-adjusted co-medication pairs were cross-referenced with known DDIs collected and curated from 15 publicly available databases (**Fig 5**). DDIs were found for 3,297 of the 3,993 co-medication pairs (83%; **S7 Table**). About half of the co-medication pairs with known DDIs involved drugs from the groups antithrombotic agents (B01), diuretics (C03), beta blockers (C07), psycholeptics (N05), and psychoanaleptics (N06), while systemic antibacterials (J01) accounted for a minuscule part of the co-medication pairs with DDI evidence (**Fig 5**, left). The proportion of co-medication pairs from the anatomical group antiinfectives for systemic use (J) not classified as DDI increased by a factor two compared to dosage-adjusted co-medication

### Drugs Used In Diabetes (A10, N=183)

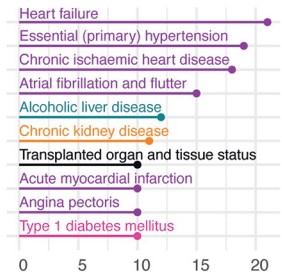
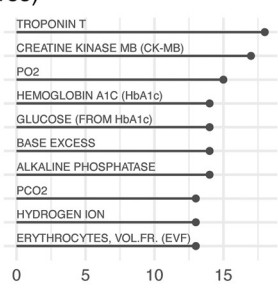
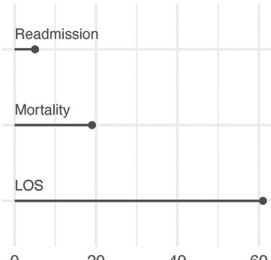

### Antineoplastic Agents (L01, N=21)

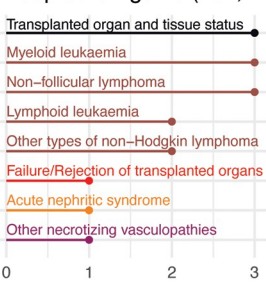
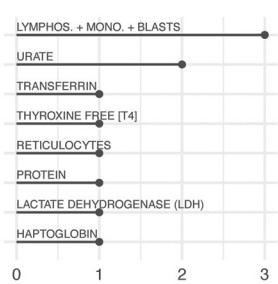
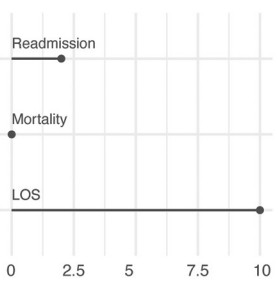

### Antithrombotic Agents (B01, N=316)

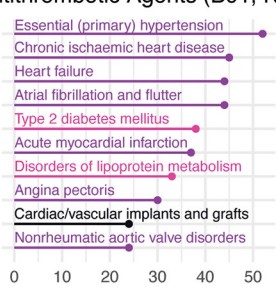
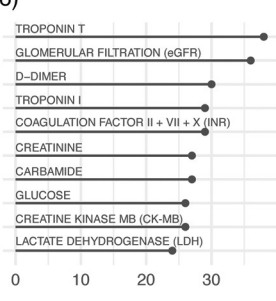
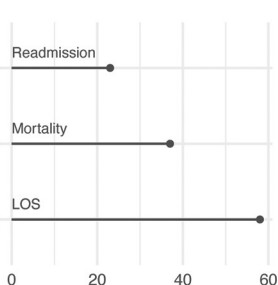

### Lipid Modifying Agents (C10, N=111)

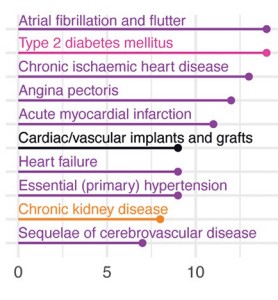
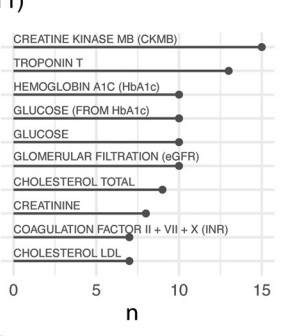
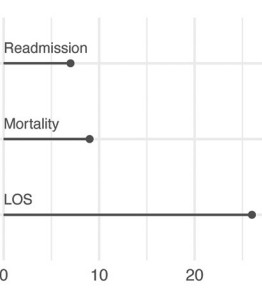

| | | |
|---|---|---|
| I — Certain infectious and parasitic diseases | VIII — Diseases of the ear and mastoid process | XV — Pregnancy, childbirth and the puerperium |
| II — Neoplasms | IX — Diseases of the circulatory system | XVI — Certain conditions originating in the perinatal period |
| III — Diseases of the blood and blood-forming organs and certain disorders involving the immune mechanism | X — Diseases of the respiratory system | XVII — Congenital malformations, deformations and chromosomal abnormalities |
| IV — Endocrine, nutritional and metabolic diseases | XI — Diseases of the digestive system | XVIII — Symptoms, signs and abnormal clinical and laboratory findings, not elsewhere classified |
| V — Mental and behavioural disorders | XII — Diseases of the skin and subcutaneous tissue | XIX — Injury, poisoning and certain other consequences of external causes |
| VI — Diseases of the nervous system | XIII — Diseases of the musculoskeletal system and connective tissue | XX — External causes of morbidity and mortality |
| VII — Diseases of the eye and adnexa | XIV — Diseases of the genitourinary system | XXI — Factors influencing health status and contact with health services |

**Fig 4. Clinical characterisation of dosage-adjusted co-medication pairs used in diabetes, cancer, and cardiovascular diseases.** For each therapeutic group, the ten most prevalent diagnoses and abnormal blood tests as well as the three outcomes (readmission, mortality and LOS) are displayed (from left to right). Segment bars indicate the number (n) of co-medication pairs positively correlated with each feature over the total pairs (N) associated with dosage adjustments for each group. LOS: length-of-stay.

pairs known as DDIs. The same trend was observed for drugs belonging to the anatomical group nervous system (N). In contrast, co-medications with drugs from the anatomical group cardiovascular system (C) comprised a larger proportion of the co-medications classified as DDIs than of the co-medication pairs that had not been classified as DDIs (**Fig 5**, right). Examples of the latter were antidiabetics (A10; e.g. cefuroxime with insulin aspart, OR:1.3, 89% High Density Interval (HDI):1.1–1.4), psychoanaleptics (N06; e.g. dicloxacillin with citalopram, OR:1.2, 89% HDI:1.1–1.4), drugs for obstructive airway diseases (R03; metronidazole with tiotropium bromide, OR:1.4, 89% HDI:1.2–1.6), and agents acting on the renin-angiotensin system (C09, e.g. gentamicin with enalapril, OR:1.3, 89% HDI:1.1–1.5). An extreme among co-medication pairs without known DDIs involved drugs for the treatment of bone diseases (M05) as co-medications (increase by a factor of four). In this case M05 co-medications involved index drugs belonging to J01 (e.g. metronidazole with alendronic acid, OR:1.4, 89% HDI:1.2–1.7), B01 (e.g. dalteparin with alendronic acid, OR:1.4, 89% HDI:1.2–1.8), N06 (e.g. mirtazapine with alendronic acid, OR:1.3, 89% HDI:1.1–1.4) and N02 (e.g. paracetamol with alendronic acid, OR:1.3, 89% HDI:1.1–1.5) (see also **S2 Table**).

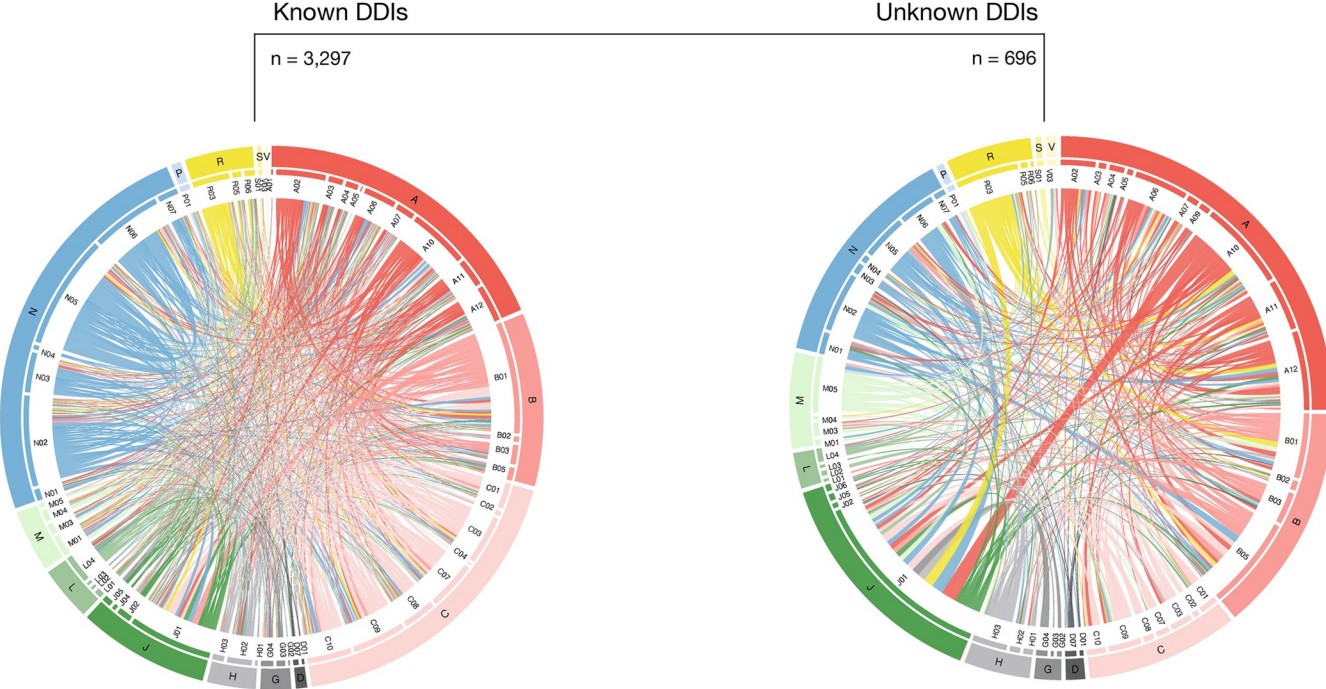

**Fig 5. Circosplots of co-medication pairs with ORs >1 with known (left) or unknown (right) DDIs.** Relations between an index drug and its co-medication are marked by connecting bands, with the width of each band proportional to the number of pairs found between two therapeutic groups (the thinnest ribbon represents one pair, e.g. C07-P01 in the set of known DDIs and A10-N04 in the set of unknown DDIs). The outer ring corresponds to the anatomical main groups, and the inner ring corresponds to the therapeutic groups. Colour of connecting bands corresponds to the anatomical group of the co-medication. See colour legend in Fig 1. The most abundant known DDI relations were observed between N05-N06, N05-N05 and B01-J01. Unknown DDIs mostly involved the therapeutic groups J01 (e.g. J01-A10), B05 (e.g. B05-N02), A12 (e.g. A12-R03), A10 (e.g. B01-A10), R03 (e.g. J01-R03) and M05 (e.g. B01-M05). DDI: drug-drug interaction. OR: odds ratio.

## Dosage adjustments, pharmacokinetics and common P450 enzyme and transporters

We investigated whether dosage-adjusted co-medication pairs had shared metabolic or transporter activity. In total, there were 1,243 co-medication pairs (31%) that were inducers, inhibitors, or substrates of overlapping CYP isozymes, 754 pairs (19%) with overlapping drug transporter and 948 (24%) with genetic variants influencing each other's metabolism/transport activity (for details see **Methods** and **S8 Table**). A total of 193 index drugs were found to have shared pharmacokinetic activities with some of their co-medications and constituted about 40% of the dosage-adjusted co-medication pairs; while 58 co-medication pairs were largely undescribed in the context of DDIs, yet with shared metabolism, transport, or pharmacogenomics.

**Fig 6** presents the 58 co-medication pairs with no DDI evidence that were inducers, inhibitors or substrates of overlapping CYP isoenzymes, transporters or annotated in PharmGKB [33] with genetic variants influencing each other's metabolism/transport activity. They primarily involved index drugs from systemic antibacterials (J01), analgesics (N02) and anaesthetics (N01) and co-medications from thyroid therapy (H03), analgesics (N02) and obstructive airway disease drugs (R03). Among these were the index drug disulfiram, with the co-medication pantoprazole which are both metabolized by CYP3A4 (OR:1.2, 89% HDI:1.1–1.4); similarly, the index drug clopidogrel, an antiplatelet drug substrate of ABCB1 transporter, with the co-medication levothyroxine sodium, an inducer of ABCB1 (OR:1.4, 89% HDI:1.2–1.6).

## Comparison of patient volumes, literature evidence and dosage adjustments

Dosage-adjusted co-medication pairs with no DDI evidence were prescribed to significantly fewer patients compared to the DDI evidence group (two-sided Mann-Whitney U-test, p-value = $6.28 \times 10^{-18}$). We searched the literature for publications mentioning the index drug and the co-medication (**S9 Table**) by full length paper text mining [34]. ORs for dosage adjustment were, indeed, significantly higher in the less prescribed and the less researched group of co-medication pairs (p-value = $2.44 \times 10^{-19}$) (**S7 Fig**). The correlation between co-mentioning and DDI evidence supports the argument that co-medication pairs with no DDI evidence are less studied (Mann-Whitney U-test, p-value = $1.5 \times 10^{-14}$). Dosage-adjusted co-medication pairs involving index drugs from the therapeutic groups antithrombotic agents (B01), antibiotics (J01) and several nervous system drug classes (N) were significantly less prescribed and with higher ORs for adjustment among the co-medication pairs with unknown DDIs compared to their corresponding pairs with DDI evidence (**S8–S10 Figs**). Although the overall differences in patient volumes, literature evidence, and dosage adjustments were small (effect size (r) = 0.137, r = 0.190, and r = 0.142, respectively), we identified large and moderate significant differences within several therapeutic groups (see **S8–S10 Figs**). For example, cefuroxime (J01DC02) and nitrazepam (N05CD02), was administered to 1,341 patients, had and OR of 1.4 and was co-mentioned in four publications (mean for J01 pairs with DDI evidence was 9,220 patients, OR:1.3 and 12 co-mentioning publications).

## Discussion

We have presented the first comprehensive, large-scale study addressing the complexity of polypharmacy at the individual drug level by means of dosage-adjustments. The study integrated drug prescription and administrative data from Danish EHRs and was designed to identify significant dosage-adjustments between specific index drugs and co-medications. We

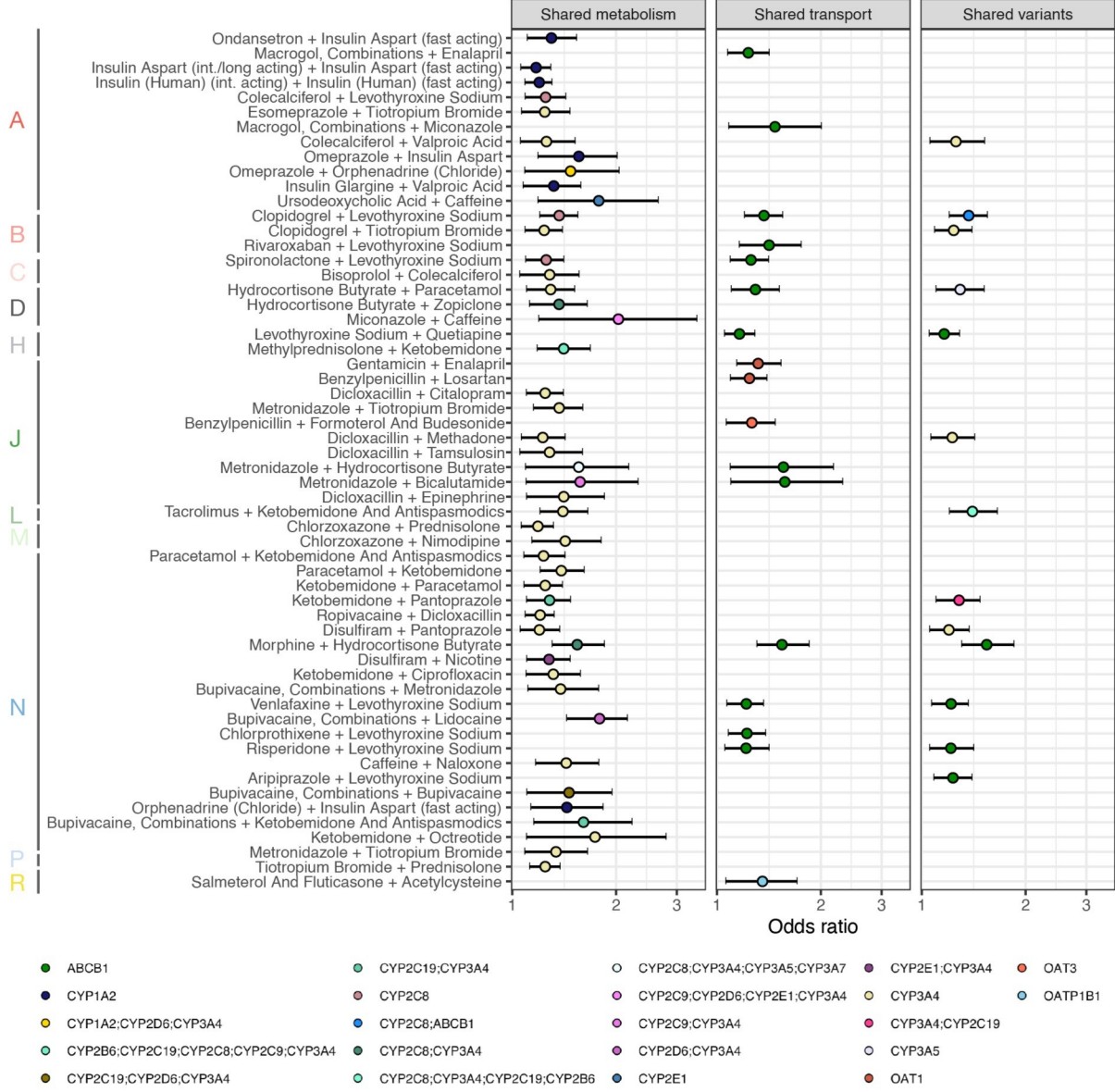

**Fig 6. Shared cytochrome isozyme and transport protein activities in dosage-adjusted co-medication pairs with no DDI evidence.**
Among the 696 co-medication pairs with ORs >1, 46 shared cytochromes, 17 shared transporters and 13 were annotated with genetic variants influencing each other's metabolism/transport. For each co-medication pair (index drug + co-medication) shown vertically the OR of dosage adjustments based on shared metabolism, transport or variants is displayed. The horizontal line indicates the Bayesian 89% high density interval (HDI). For each anatomical group of the index drug, pairs were ordered according to their prevalence (top, highest).

identified 77,494 co-medication pairs of which 3,993 were likely to be dosage-adjusted. Our analysis includes 1.1 million in-patients and more than 184 million treatment episodes.

## Clinical impact of co-medications pairs and DDIs

By linking the dosage-adjusted co-medication pairs to clinical outcomes, diagnoses, and blood test results we display their clinically meaningful impact. The fact that dosage-adjusted co-medication pairs that associated with readmission and mortality were dominated by the anatomical group nervous system (N) and antibiotics for systemic use (J01) adds to previous

studies of potentially severe drug interactions. Our study contributes to the existing literature by presenting a comprehensive analysis of co-medications administered concomitantly in an in-hospital setting, surpassing the number of co-medications reported in previous studies. For instance, while Dumbreck et al. characterized 243 DDIs based on pre-defined phenotypes [35], our study expands this by providing a larger dataset and extending the scope of analysis. One obvious advantage of this approach is the possibility to discover potential unknown DDIs, which previous studies were short of. The 694 dosage-adjusted co-medication pairs that did not classify as known DDIs hold significant clinical interest and may be of utter importance in future studies aimed at addressing the current limited applicability of clinical guidelines to patients with comorbid conditions [9,35].

Further, we have exemplified that the 3,993 co-medication pairs add valuable information in the context of a wide variety of other healthcare data. For example, anti-infective drugs (J) were generally underrepresented in DDI databases, although they are among the most prevalently prescribed drugs; and as our study demonstrates were often dosage-adjusted when co-administered with other drugs. Similarly, the anti-infective drugs were only included in the "total number of medications" in the Drug Burden Index developed by Hilmer et al. [11]. As our results demonstrate, a simple accumulation of prescribed drugs does not capture the complexity of polypharmacy. Of the three outcomes, antithrombotics (B01) did generally not associate with LOS, whereas many antibiotics for systemic use (J01) associated with both readmission and mortality. These findings further highlight the need for developing interdisciplinary guidelines that account for the fact that some drugs are administered to nearly all types of patients [2].

## Co-medication pairs as a steppingstone to map mechanisms of DDIs

Ultimately, genetic information will support a better understanding of the great variability in patient response to medication. In this study, we succeeded in characterizing all drug pairs administered concomitantly in an in-hospital setting over a period of eight years. A total of 1,525 dosage-adjusted co-medication pairs were inducers, inhibitors, or substrates of overlapping CYP isoenzymes, drug transporters, or otherwise influencing the same pharmacokinetic pathways. Thus, our study may compliment ongoing efforts focusing on dosage variability in relation to patient genotypes. As Wu and colleagues demonstrated that the maintenance dose correlated with patient genotypes [21], it would be interesting to stratify dosage-adjusted co-medication pairs by patient genotype to further characterize them. Interestingly, they find two medication-associated genes in two medication classes that have not previously been subjected to GWAS, which overlap with our findings: thyroid therapy (H03) and drugs for acid related disorders (A02; e.g., pantoprazole), which were prevalent among the 58 co-medication pairs with no DDI evidence (**Fig 6**).

As dosage-adjusted co-medications have not previously been described in the context of DDIs and pharmacokinetics, we suggest that these pairs provide a basis for developing a strategy to identify potentially novel DDIs. For example, methotrexate has previously been linked to alterations on electrolyte handling and increased glucose uptake [36,37]. We find that methotrexate appears more likely adjusted with electrolyte and carbohydrate co-medications. Similarly, many antibiotics (i.e. flucloxacillin, gentamicin, erythromycin, cefuroxime, dicloxacillin, meropenem and piperacillin) were adjusted with the co-medication insulin aspart, which may reflect the altered metabolic control during infection in patients with diabetes [38]. In sum, the results of our study have potential for further analysis focusing on the mechanisms that warrant the necessity of dosage-adjustments. Moreover, our findings may also guide future studies of drug response pharmacogenetics, where it is often necessary to prespecify drugs and genes of interest [23].

### The value of electronic health records

In contrast to previous studies focussing on polypharmacy broadly, the data foundation of our study is structured EHRs, whereas the studies by Wu et al. [21] and Aoki et al. [22] have relied on survey data and questionnaires. We argue that this aspect of our study is an inherent strength as it enables us to assess polypharmacy in a population-wide manner. However, that does no eliminate the limitations, which come with an observational study. Thus, the results do not allow us to draw any causal conclusions. However, we find that for 1,949 (49%) of dosage-adjusted co-medications pairs, there was a high likelihood of the index drug being discontinued during concomitant treatment. This observation supports the general interest in the field of drug discontinuation and generalized the scope of previous studies focusing on e.g., terminal cancer patients [13]. As polypharmacy presents an additional risk to multimorbid patients, a focus on drug-discontinuation could be an interesting future analysis.

There are also other limitations. For example, while we adjusted for sex, age, diagnosis at admission, hospital, calendar year, and patient heterogeneity, factors such as weight, kidney and liver function and other patient characteristics i.e., reduced mobility, delirium, hypoglycaemia, and high blood pressure that may account for dosage changes were not considered. Also, the use of inpatient data may relate to a sicker population with a higher degree of complications. Yet, the use of a large inpatient cohort over an outpatient cohort, allowed us to control for and avoid the well-known lack of adherence associated with the latter, and which would have rendered it difficult to track dosage changes reliably [39]. As only in-patient data was used, we cannot exclude that co-medications prescribed before admission could have an effect. Similarly, the starting time of concomitant drugs is not considered primarily because co-medications that started before admission may not be accurately recorded. Future endeavours could include different pharmacological aspects of compartmentalization, pharmacokinetics, and pharmacodynamics as well. Finally, drug pairs per se present a way of simplifying a complex problem of polypharmacy. While dosage-adjustments in the context of polypharmacy is often overlooked, we do not succeed in disentangling the contribution of each drug to the full drug combination profile as analysing more than two drugs simultaneously would require of a systematic assessment over all such sub-combinations [40]. Given the data set size (that is already large) this would lead to a rather non-uniform sampling and a more anecdotal analysis. Another caveat of simplification as a mean to obtain a comprehensive analysis is that we did not find an association between warfarin and ibuprofen or between the macrolide erythromycin and the CYP3A4-inhibitor verapamil, although DDIs are well-characterized for these drugs. In the study we used the DDI information in a binary form, present or non-present, and this is obviously a simplification. In sum, we argue that the strengths of the study by far outweighs the limitations and that Bayesian inference is an important methodology for studies within this field.

## Conclusion

Overall, this study introduces a novel method that specifies and quantifies the level of polypharmacy derived from real-world data. In a cohort of 1,069,873 in-patients, we identified 77,249 co-medications pairs of which 3,993 were more likely to be dosage-adjusted when co-administered with other drugs. The study serves as an important platform for the understanding of all sources of drug response variability, independently and systematically; which will then allow for better assessments of drug efficacy and for limiting the impact of the increasing polypharmacy patients experience today.

## Methods

### Ethics statement

This study was approved by the Danish Patient Safety Authority (3-3013-1731 and 3–3013–1723), the Danish Data Protection Agency (DT SUND 2016–48, 2016–50, 2017–57 and UCPH 514-0255/18-3000) and the Danish Health Data Authority (FSEID 00003092, FSEID 00003724, FSEID 00004758 and FSEID 00005191). Reporting of results follows the Strengthening the Reporting of Observational Studies in Epidemiology (STROBE) reporting guideline [41] (**S10 Table**).

### Study design

The data used in this study was obtained from EHRs of 12 public hospitals in the Capital Region and Region Zealand of Denmark covering the period January 2008 to June 2016. We only considered inpatient hospital admissions, which included 3,161,647 million records for 1,069,873 million individuals. Admissions covering two or more discharges between wards were combined into one. Similarly, in cases of re-admission within 24 hours of discharge, records were also combined. The primary diagnosis of each admission recorded at discharge was coded according to the International Statistical Classification of Diseases and Related Health problems 10[th] Revision (ICD-10). From the EHRs, two different medication modules were used to retrieve in-hospital drug prescription data: Electronic Patient Medication (EPM) and OPUS-medication (OpusMed). The former has been previously validated as a useful tool for pharmacoepidemiologic research [42], and the latter has been used in a similar manner making it a credible source, too. The modules contain time-stamped data on prescribed drugs and instructions regarding the dosage regimen. All drugs were coded according to the Anatomical Therapeutic Chemical Classification (ATC).

### Calculation of prescribed dosage

For all drug prescriptions, average daily doses (ADDs) were computed. ADD was calculated according to the dosage regimen indicated in the drug prescription. ADD represents the clinical decision of the treatment plan regarding the prescribed dose, frequency, and interval (Eq 1). Drug dose ($d$) refers to the dose for each single administration; dose frequency ($f$) represents the number of times the dose needs to be taken in the dose interval. Dose interval ($t$) refers to the time interval in hours to which the dose frequency applies. The term $n$ represents the number of different drug doses ($d$) and frequencies ($f$) that may be defined during a prescribed interval ($t$) (**S11 Fig**).

$$ADD = \frac{1}{t} \sum_{i=1}^{n} d_i \cdot f_i \tag{1}$$

ADD was calculated for all prescriptions containing information for the prescribed dose, frequency, and interval of administrations. ADD of medications to be taken as needed and with variable dose were coded as categorical ADD. In cases where prescribed dose was not specified, the ADD was calculated given the strength indicated in the drug package label information and the prescribed volume. If the prescribed volume was not specified, we assumed 1 unit of volume. If the dose interval was not indicated, but the frequency was specified, the interval was derived from it. If the interval and frequency were not specified, we assumed a daily interval. Finally, if frequency was not defined, we assumed 1 administration per interval. Prescriptions for whose ADD could not be calculated represented less than 10% of the total prescriptions analysed (n = 1,790,106) and were removed.

## Construction of concomitant treatment episodes

Treatment episodes were defined as the events from the start to the end of the administration of a medication [43–48]. We constructed concomitant treatment episodes as the intervals of time when two different drug prescriptions were contemporaneously active. We also constructed a reference treatment episode as the intervals of time when a drug was not concomitantly given with any other drug ('monotherapy episode') (**S12 Fig**). Concomitant treatment episodes were constructed in a temporal manner accounting for all the modifications in the prescribed regimen of a medication in the drug profile of every single individual, for both concomitant and monotherapy episodes. Treatment adjustments were defined as the modification on the prescribed dosage (ADD) during different episodes. Treatment episodes for which complete sequence of prescribed ADD could not be calculated were excluded from the analysis.

Treatment episodes were included in regressions that estimated likelihoods for dosage adjustments during concomitant treatment episodes. Each co-medication pair was then characterized by the likelihood of dosage adjustment for the index drug using odd ratios (ORs). The monotherapy episodes of the corresponding index drug were used as reference.

## Bayesian inference

The objective of the analysis was to estimate the probability of adjusting the dosage of a treatment associated to the simultaneous exposure with other co-medications. A pertinent flowchart for the selection criteria can be found in **S13 Fig**. We examined all drug pairs present in 50 patients or more and all drugs that had a monotherapy episode in at least 50 patients. We also excluded 77 drugs that were prescribed as one-time drugs in more than 70% of their prescriptions only when these were considered as co-medications (but not for index drugs) (**S11 Table**).

Dosage adjustments were modelled using a series of hierarchical Bayesian logistic regression models. For each individual episode, the total number of modifications $y$ was calculated as the number of times there was a lagged change in the prescribed dosage, where $N$ is the number of prescriptions during an episode:

$$y = \sum_{i=1}^{N}(ADD_{i+1} - ADD_i) \tag{2}$$

The total number of prescriptions during a concomitant treatment episode $N$ defined the total number of trials. We then defined the probability mass function of the binomial distribution as:

$$P(y|N,p) = \binom{N}{y}p^y(1-p)^{N-y} \tag{3}$$

where $p$ is the event probability. In the binomial model, we directly predict $p$ at the logit-scale, which means that for each observation $i$ we compute the success probability of a dosage change $p_i$ as:

$$logit(p_i) = \log\left(\frac{p_i}{1-p_i}\right) \tag{4}$$

where $p_i$ is the linear combination of the subset of concomitant and monotherapy episodes for

every drug formally expressed by:

$$p_i = \beta_{i,0} + \beta_{i,concomitant\ drug} + \beta_{i,age} + \beta_{i,sex} + \beta_{i,hospital} + \beta_{i,diagnosis} + \beta_{i,year} + \gamma_{i,patient} \qquad (5)$$

We included age, sex, hospital, calendar year, primary admission diagnosis at the chapter level and a term for each concomitant drug (co-medication) as fixed effects in the model. We corrected for the main admission diagnosis as different admission reasons presented significantly different medication burdens (see **S1 Table**). The criteria for including hospitalisation year accounted for changes in the market due to drug approvals, withdrawals and modification of guidelines that may influence the analysis [49,50], such as with the decreasing trends of prescription in the case of opioids and antibiotics [51,52]. To account for unmeasured heterogeneity correlated with individuals and multiple visits, a patient random effect was included in the model.

To complete the model, we specified a set of priors on the coefficients:

$$\beta_{i,0} \sim N(0, \sigma_0) \qquad (6)$$

$$\beta_{i,concomitant\ drug} \sim N(0, \sigma_{concomitant\ drug}) \qquad (7)$$

$$\beta_{i,age} \sim N(0, \sigma_{age}) \qquad (8)$$

$$\beta_{i,sex} \sim N(0, \sigma_{sex}) \qquad (9)$$

$$\beta_{i,hospital} \sim N(0, \sigma_{hospital}) \qquad (10)$$

$$\beta_{i,diagnosis} \sim N(0, \sigma_{diagnosis}) \qquad (11)$$

$$\beta_{i,year} \sim N(0, \sigma_{year}) \qquad (12)$$

$$\gamma_{i,patient} \sim N(0, \sigma_{patient}) \qquad (13)$$

We assigned weakly informative priors to all coefficients, i.e. a normal distribution with zero mean and a variance of 1. Note that, the adjective "weakly informative" prior used in this study is the classical wording but does not necessarily mean that the prior is truly non-informative. The random effect was also assumed to follow a normal distribution and the standard deviation of the random effects is given a normal prior distribution between 0 and 1. We defined the prior on the scales of the coefficients as shown in Eqs [14–20].

$$\sigma_{concomitant\ drug} \sim N_+(0, 1) \qquad (14)$$

$$\sigma_{age} \sim N_+(0, 1) \qquad (15)$$

$$\sigma_{sex} \sim N_+(0, 0.5) \qquad (16)$$

$$\sigma_{hospital} \sim N_+(0, 1) \qquad (17)$$

$$\sigma_{diagnosis} \sim N_+(0, 1) \qquad (18)$$

$$\sigma_{year} \sim N_+(0, 1) \tag{19}$$

$$\sigma_{patient} \sim N_+(0, 1) \tag{20}$$

The models were fitted and posterior distributions were estimated using the R package *brms* (v2.8.0) [53]. A total of four chains for 2,000 iterations each were run, with a burn-in period of 1,000 iterations per chain. Model convergence was assessed by inspecting divergences, tree depth and R-hat values, which indicate if the four chains arrived at approximately the same posterior distribution for the parameters [54]. We concluded that models converged with R-hat values below or equal to 1.1, tree depths not at maximal in any of the chains after warm-up, and zero divergences. For computational modelling reasons, drugs with more than 500,000 concomitant episode observations (n = 88) were down sampled following a stratified approach retaining the distribution of the original set of observations. Down-sampling was done by first selecting the most frequent 300 co-medications, and then reducing the number of observations to 500,000 episodes randomly selected based on the frequency distribution of six variables: age, sex, calendar year, hospital, admission reason and concomitant drug. The stratified sampling was performed by computing probability intervals for all six variables iteratively on the filtered data set. Probability intervals for categorical variables were based on the frequency of their values in the population, and age was discretized based on Rice's rule criterion (Eq 21) [55]. Co-medications that after down-sampling were prescribed to less than 50 patients were excluded from the subsequent analysis.

$$k = \lceil 2n^{1/3} \rceil + 1 \tag{21}$$

## Adjusted odds ratio of dosage modifications

Adjusted odds ratios (ORs) for each co-medication pair were reported, using treatment episodes without any concomitant drugs ('monotherapy') as a reference. Posterior distributions were summarized with 89% High Density Interval (HDI) [56] of the OR. HDI is the portion of the posterior distribution containing the percentage of probable values. HDI with 89% intervals were used as these are considered to be more stable than 95% intervals [56] as several effective sample sizes (ESS) from the posterior distribution were not always >200 or >1000. An ESS of 10,000 is recommended for choosing the 95% interval [57]. ORs can be interpreted as the likelihood the administration with a co-medication was observed with a change in the dosage in the following prescription of the index drug. The reported OR is the median of the posterior distribution. In order to assume clinically meaningful associations, we defined a Region Of Equivalent Practice (ROPE) in the interval (-0.05, 0.05) equivalent to a null region. If the HDI is completely outside the ROPE, the null hypothesis was rejected; whereas if all most values of HDI covers the ROPE the null hypothesis was accepted [58].

## Clinical characterisation of the co-medication pairs

To characterize co-medication pairs with respect to the clinical context, we analysed the association of all significant dosage-adjusted co-medication pairs (n = 3,993) in three types of analyses. First, we analysed the association of co-medication pairs with discontinuations using a logistic regression with exposure to the co-medication drug as the main exploratory variable. Second, we analysed the association with different outcome measures: 30-days post-admission all-cause mortality rate, 30-days post-discharge readmission rate and LOS. Mortality and readmission were analysed using stratified Cox regression models assuming non-informative

censoring, and LOS was analysed using stratified Poisson regression with exposure to the co-medication pair as the main explanatory variable. Last, we analysed the association of all significant co-medication pairs with blood tests and diagnoses during admission. Blood tests were evaluated as binary outcomes (within reference range or not) and similarly for diagnoses (assigned or not assigned). If relevant, the reference ranges differ for different age groups etc. Missing values (blood tests not taken) were considered as normal within range results (rationale: if the test was not taken, there was no strong suspicion of altered physiology). We considered diagnoses and tests at least present in 50 patients. The enrichment of diagnoses and biochemical tests was evaluated using logistic regression between exposed and non-exposed groups, where the dependent variables were abnormal biochemical tests or assigned diagnosis codes, respectively.

In the three post-hoc analyses, patients who had been prescribed a dosage-adjusted co-medication pair were considered exposed. Exposed patients were matched to a non-exposed control group that received the same index drug but a different concomitant drug by 1:1 propensity score matching. We used the R package *MatchIt* (v4.0.0) [59] and performed a nearest neighbour matching comparison using the propensity score difference as the distance metric [60]. To account for potential biases, exposed and non-exposed groups were also matched based on age, sex, primary diagnosis (ICD-10 chapter level), weighted Charlson comorbidity score [61] and medication burden (number of unique drugs during the admission). Only patients' first exposure to any co-medication pair was used. We adjusted all models for age, sex, primary diagnosis, weighted Charlson score and medication burden. P-values were adjusted using Bonferroni correction for multiple testing, where a p-value $\leq 0.05$ was considered significant.

## Drug-drug interaction data

To analyse the overlap between co-medication pairs associated with dosage changes and DDIs, drug interaction data from the 15 most widespread DDI databases were collected following a similar approach to Ayvaz et al. [62]. These were DrugBank [63], Danish Medicines Agency database for drug interactions [64], NLM CV DDI Corpus [62], ONC-HP [65], ONC-NI [66], Corpus PK [67], Corpus 2013 [68], Corpus 2011 [69], Twosides [70], CredibleMeds [71], KEGG Medicus [72], VA NDF-RT, HEP Drug Interactions database (www.hep-druginteractions.org), HIV Drug Interactions database (www.hiv-druginteractions.org) and Cancer Drug Interactions database (www.cancer-druginteractions.org). The collection includes eight clinically oriented information sources (Cancer, HIV, HEP, ONC-NI, the official Danish drug interaction database and VA NDF-RT), three bioinformatics/pharmacovigilance resources (DrugBank, Twosides and KEGG) and four natural language processing (NLP) corpora. In total, the collection contains 2,558,568 unique DDIs. Across the data sources there were little overlap (**S14 Fig**). The low consistency and overlap among databases could be explained by different focus areas i.e. on specific disease (i.e. HIV, HEP, Cancer) or complications (CredibleMeds and QT intervals) depicting the nature or purpose for which they were generated or from where they were generated (**S15 Fig**).

## Drug metabolism and transport data

Enzyme metabolism and protein transport data was retrieved from DrugBank [63].

We limited the investigation to a common set of cytochromes and drug transporters recommended to be evaluated by regulatory agencies: CYP1A2, CYP2B6, CYP2C8, CYP2C9, CYP2C19, CYP2D6, CYP2E1, CYP3A4, CYP3A5, CYP3A7, ABCB1, BCRP, MATE1, Mate-2k, OAT1, OAT3, OCT1, OCT2, OATP1B1, OATP1B3 and P-SP [16,17].

We assumed a common metabolism or transport activity if index drug and co-medication were inducers, inhibitors, or substrates for the same CYP enzyme or transport protein.

Information regarding variant annotations of associated genes with drugs was retrieved from PharmGKB [33]. We assumed a drug and its co-medication as potential DDGI when an associated gene variant could interfere with each other's metabolic or transport activity, either by induction, inhibition, or substrate, or with an associated gene variant associated to a CYP or a transport protein.

### Literature data

Literature search and statistics about co-mentioned concomitant drugs were extracted using a previously described full-text article corpus comprising up to 15 million full text articles as well as 16.5 million additional abstracts in PubMed and MEDLINE [34]. This corpus was reduced to only include articles indexed in MEDLINE, as these are the ones including journals related to Health Sciences, Chemistry and Life Sciences, and therefore the ones most likely covering DDIs. We used the following rules to decide which texts to include: (1) use the full text for an abstract if available from PubMed Central; (2) use the full text for an abstract if available from the full text corpus; (3) otherwise, use the abstract (**S16 Fig**). In total we analysed 3.8 million articles of which the full text was available (1.1 million non-PubMed Central and 2.7 million from PubMed Central) as well as 27.7 million MEDLINE abstracts.

### Comparison of patient volumes, literature evidence and dosage adjustments

Because the patient volumes, DDI classifications, and dosage adjustments were all non-normally distributed (p-value < 0.05 using Shapiro-Wilk test of normality), the two-sided Mann–Whitney U-test was used compare dosage-adjusted co-medication pairs with and without DDI evidence. Specifically, we tested if the number of patients, number of co-mentions and ORs were significantly higher in co-medication pairs with known DDI evidence compared to co-medication pairs with unknown DDI evidence. The "wilcox_effsize" function from the R package *rstatix* (v0.7.2) was used to calculate the effect size of the differences.

### Supporting information

**S1 Fig. Polypharmacy and age.** Age distribution based on the number of concomitant drugs (1, 2–4, 5–9, ≥10). Pearson $\rho$:0.40, 95% confidence interval CI:0.39–0.40, p-value < 2.2x10-16. CI: Confidence Interval.
(TIF)

**S2 Fig. Conceptual overview for drug co-administration and dosage patterns. a** Example of an inpatient admission where three index drugs (drug A, drug B and drug C) are administered. Timestamped prescriptions (Rx n) are shown for each drug (four for drug A, three for drug B and one for drug C), the colour grading indicates the prescribed average daily dose (ADD). **b** For each index drug, concomitant treatment episodes are defined as the co-medication overlap between a pair of drugs. The dotted area exemplifies the concomitant episodes for the index drug A with co-medication drug B, A(B). During A(B), the index drug A is added, followed by three consecutive prescriptions (Rx1, Rx2, Rx3) and then discontinued (D/C).
(TIF)

**S3 Fig. Characterisation of dosage patterns and co-medications.** Stacked barplots representing the proportion of co-medications correlated with co-medications for each index drug. The index drugs with the highest proportion of co-medications with ORs >1 were: piperacillin

(J01CR05), cefuroxime (J01DC02), caffeine (N06BC01), insulin glargine (A10AE04), methotrexate (L01BA01) and atorvastatin (C10AA05), consisting of more than 18% of all their co-medications. OR: Odds Ratio.
(TIF)

**S4 Fig. Odds ratio for dosage adjustment with co-medications.** Representation of the 309 index drugs with dosage adjustments having ORs >1 with co-medications. Each point represents the OR of a co-medication for the given index drug. OR: Odds Ratio.
(TIF)

**S5 Fig. Top 20 diagnoses correlated with dosage adjusted co-mediation pairs.** Segment bars indicate the number (n) of co-medication pairs positively correlated with each diagnosis over the total 3,993 dosage adjusted co-medication pairs. Colour of the bars correspond to the chapter level of the ICD-10 classification of diseases.
(TIF)

**S6 Fig. Top 20 abnormal laboratory tests correlated with dosage adjusted co-medication pairs.** Segment bars indicate the number (n) of co-medication pairs positively correlated with each laboratory test over the total 3,993 dosage adjusted co-medication pairs.
(TIF)

**S7 Fig. Comparison between patient volume, DDI evidence and dosage adjustments. a** Boxplots indicating the number of patients and the DDI evidence among co-medication pairs with ORs >1 (Two-sample Mann-Whitney U-test, p-value = $6.28 \times 10^{-18}$); **b** Boxplots of the number of publications where co-medication pairs appeared co-mentioned (p-value = $1.52 \times 10^{-14}$); **c** Boxplots of the odds ratio for dosage (p-value = $2.44 \times 10^{-19}$). OR: Odds Ratio.
(TIF)

**S8 Fig. Comparison between dosage adjustments and DDI evidence.** Boxplots indicating the OR and the DDI evidence among dosage adjusted co-medication pairs across the different index therapeutic drug groups. All significant differences showed small effect sizes, except vitamins (A11) with a moderate effect size. Two-sample Mann-Whitney U-test, where *: p-value $\leq$ 0.05, **: p-value $\leq$ 0.01, ***: p-value $\leq$ 0.001, ****: p-value $\leq$ 0.0001. Therapeutic groups with less than five observations for each DDI evidence class (DDI/Unknown DDI) are not shown. OR: Odds ratio.
(TIF)

**S9 Fig. Comparison between patient volume and DDI evidence.** Boxplots indicating the prevalence (number of patients) and the DDI evidence among dosage adjusted co-medication pairs across the different index therapeutic drug groups. All significant differences showed small effect sizes, except anesthetics (N01) and antiemetics and antinauseants (A04) with a moderate effect size. Two-sample Mann-Whitney U-test, where *: p-value $\leq$ 0.05, **: p-value $\leq$ 0.01, ***: p-value $\leq$ 0.001, ****: p-value $\leq$ 0.0001. Therapeutic groups with less than five observations for each DDI class (DDI/Unknown DDI) are not shown. OR: Odds Ratio.
(TIF)

**S10 Fig. Comparison between literature and DDI evidence.** Boxplots indicating the number of publications with co-mentioning and the DDI evidence among dosage adjusted co-medication pairs across the different index therapeutic drug groups. All significant differences showed small effect sizes, except blood substitutes and perfusion solutions (B05), antibiotics (J01) and antiemetics and antinauseants (A04) with a large effect size; and mineral supplements (A12),

vitamins (A11), drugs for functional gastrointestinal disorders (A03), analgesics (N02), antidi-arrheals and antiiflammatory/antiinfective agents (A07), drugs for obstructive airway diseases (R03) and anesthetics (N01) with a moderate effect size. Two-sample Mann-Whitney U-test, where *: p-value $\leq$ 0.05, **: p-value $\leq$ 0.01, ***: p-value $\leq$ 0.001, ****: p-value $\leq$ 0.0001. Therapeutic groups with less than five observations for each DDI class (DDI/Unknown DDI) are not shown. OR: Odds Ratio.
(TIF)

**S11 Fig. Average daily dose.** Calculation of prescribed dosage as the prescribed average daily dose (ADD). A prescription is a health-care program implemented by a physician or other qualified health care practitioner in the form of instructions that govern the plan of care for an individual patient. The term often refers to a health care provider's written authorization for a patient to purchase a prescription drug from a pharmacist. Prescriptions can be classified into four different groups: (1) one-time prescriptions; (2) scheduled prescriptions; (3) Pro Necessi-tata (PN) or 'as needed' prescriptions; and (4) Variable dosage (VAO) prescriptions. One-time prescriptions consist on the administration of a single dose in a single day. As this type of pre-scriptions do not allow the possibility to study longitudinal dosage adjustments, we did not consider them for this study. PN consist of drugs only administered if needed and they are indicated with a maximum number of administrations/day that can be taken. An example of a PN prescription is the painkiller paracetamol or other analgesics indicated for the treatment of pain if the patient requires so after e.g. surgery. VAO drugs refer to drugs whose dosage is vari-able on a biochemical value or another physiological constant of the patient at the time of each administration. For example, this is the case for insulin, whose dosage depends on the blood glucose levels. For PN and VAO drug prescriptions, the ADD was coded categorically as 'PN' or 'VAO'. Henceforth, quantitative ADD was calculated only for scheduled prescription types. Scheduled prescriptions consist of drug indications having an interval or cycle with a defined time pattern for each administration. Among this type, we can find many variations with dif-ferent intervals and patterns. The figures **a** and **b** exemplify two different examples: **a** Dosing regimen for acyclovir (J05AB01), a drug used to treat infections caused by certain types of viruses (e.g. cold sores, shingles, chickenpox). The prescription regimen is indicated with a daily interval (1 day), a frequency of three administrations following the time pattern at 8 am, 2 pm and 10 pm. The prescribed dose indicated at each administration is of 200 mg. The final calculated ADD for acyclovir is of 600 mg (see Eq 1). **b** A more complex regimen is presented with the medication Levothyroxine sodium (H03AA01), a thyroid hormone medication that is used to treat hypothyroidism and other hormonal conditions. The treatment plan for this drug follows a weekly interval (7 days) and different prescribed doses for every other day. The pre-scribed dose is 50 micrograms (mcg) for days 1, 3, 5 and 100 mcg for days 0, 2, 4, 6. The final calculated ADD is 78.57 mcg, which is the sum of the two previous prescribed doses in the same weekly interval.
(TIF)

**S12 Fig. Construction of concomitant and monotherapy treatment episodes.** Patients can have a drug prescribed multiple times during an admission. We constructed concomitant treatment episodes as the intervals of time when two different drug prescriptions were contem-poraneously active from start to end of the administration of a medication. Monotherapy treatment episodes were used as the reference treatment episodes as the intervals of time when a drug was not concomitantly given with any other drug.
(TIF)

**S13 Fig. Flowchart for the reduction of the initial 1,539 drugs that were prescribed during the study period to 413 drugs fulfilling the selection criteria.**
(TIF)

**S14 Fig. Overlap among DDI data sources.** Upset plot showing the overlap of DDIs among different DDI source types: bioinformatics/pharmacovigilance, clinically oriented and Natural Language Processing (NLP) corpora. The upset plot is conceptually like a Venn diagram. The vertical histogram on the right shows the number of overlapping DDIs and the dots underneath show the source types containing them. The greatest overlap was of 122,454 DDIs between clinically oriented and bioinformatics/pharmacovigilance sources. DDI: drug-drug interaction.
(TIF)

**S15 Fig. Overview of DDI data sources.** Proportion and relation between DDIs of different drug classes across the different databases used. Databases are sorted from left to right, top to bottom, according to the number of DDIs they contain (From DrugBank, the highest, to the lowest, CredibleMeds). Relation between drug classes is indicated by connecting bands, where the width is proportional to the number of pairs. The outer ring corresponds to the anatomical main group of the Anatomical Therapeutic Chemical (ATC) classification system. ONC-HP: ONC High Priority, ONC-NI: ONC Non-Interruptive, HEP: Hepatitis DDI database, HIV: Human Immunodeficiency Virus DDI database.
(TIF)

**S16 Fig. Literature co-mentions diagram.** Diagram of the full-text article and abstract corpus generation for the text mining of DDIs. DTU DTM Corpus (~15 million articles), PubMed (~27 million articles) and MEDLINE were used for the collection of full-text articles and abstracts.
(TIF)

**S1 Table. Admission primary diagnosis.**
(TSV)

**S2 Table. Co-medication pairs with associated dosage adjustment.** List of co-medication pairs with ORs >1 for dosage adjustment. OR: Odds Ratio.
(TSV)

**S3 Table. Dosage-adjusted co-medication pairs with associated discontinuation of treatment.** List of co-medication pairs with ORs >1 for dosage adjustment and their association with discontinuation (cessation of treatment). OR: Odds Ratio.
(TSV)

**S4 Table. Dosage-adjusted co-medication pairs with associated outcomes.** List of co-medication pairs with ORs >1 for dosage adjustment and their association with outcomes (readmission, mortality, LOS). OR: Odds Ratio. LOS: Length-of-stay.
(TSV)

**S5 Table. Dosage-adjusted co-medication pairs with associated diagnoses.** List of co-medication pairs with ORs >1 for dosage adjustment and their association with admission diagnoses. OR: Odds Ratio.
(TSV)

**S6 Table. Dosage-adjusted co-medication pairs with associated abnormal biochemical test results.** List of co-medication pairs with ORs >1 for dosage adjustment and their association

with abnormal biochemical test results. OR: Odds Ratio.
(TSV)

**S7 Table. DDI evidence.** List of co-medication pairs with ORs >1 for dosage adjustment and their DDI evidence. OR: Odds Ratio.
(TSV)

**S8 Table. Shared CYPs and transporters.** List of co-medication pairs with ORs >1 for dosage adjustment and their pharmacokinetic activities: shared metabolism and transport, and associated variants. OR: Odds Ratio.
(TSV)

**S9 Table. Literature co-mentions.** List of co-medication pairs with ORs >1 for dosage adjustment, DDI evidence and co-mentioning in the literature. OR: Odds Ratio.
(TSV)

**S10 Table. STROBE checklist.**
(DOCX)

**S11 Table. One-time drugs.** List of excluded index drugs for which the prescription type was 'one-time' or 'single-administration' in more than 70% of the cases.
(TSV)

## Author Contributions

**Conceptualization:** Cristina Leal Rodríguez, Søren Brunak.

**Data curation:** Cristina Leal Rodríguez.

**Formal analysis:** Cristina Leal Rodríguez, Gianluca Mazzoni.

**Funding acquisition:** Søren Brunak.

**Investigation:** Amalie Dahl Haue, Robert Eriksson.

**Methodology:** Cristina Leal Rodríguez, David Westergaard, Søren Brunak.

**Project administration:** Cristina Leal Rodríguez, Søren Brunak.

**Resources:** Cristina Leal Rodríguez, Robert Eriksson, Jorge Hernansanz Biel, Kirstine G. Belling, Søren Brunak.

**Supervision:** Søren Brunak.

**Validation:** Lisa Cantwell.

**Visualization:** Cristina Leal Rodríguez, Amalie Dahl Haue, Søren Brunak.

**Writing – original draft:** Cristina Leal Rodríguez, Amalie Dahl Haue, Søren Brunak.

**Writing – review & editing:** Cristina Leal Rodríguez, Amalie Dahl Haue, Gianluca Mazzoni, Robert Eriksson, Jorge Hernansanz Biel, Lisa Cantwell, David Westergaard, Kirstine G. Belling, Søren Brunak.

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
