## [Decision Letter · Decision Letter 0]

1 Jun 2023

PDIG-D-23-00017

Drug dosage modifications in 24 million in-patient prescriptions covering eight years:  A Danish population-wide study of polypharmacy

PLOS Digital Health

Dear Dr. Brunak,

Thank you for submitting your manuscript to PLOS Digital Health. After careful consideration, we feel that it has merit but does not fully meet PLOS Digital Health's publication criteria as it currently stands. Therefore, we invite you to submit a revised version of the manuscript that addresses the points raised during the review process.

Please submit your revised manuscript within 30 days Jul 01 2023 11:59PM. If you will need more time than this to complete your revisions, please reply to this message or contact the journal office at digitalhealth@plos.org. Please include the following items when submitting your revised manuscript:

We look forward to receiving your revised manuscript.

Kind regards,

Benjamin P. Geisler, M.D., M.P.H., F.A.C.P., M.R.C.P. (London), F.H.M.

Academic Editor

PLOS Digital Health

Journal Requirements:

2. Please send a completed 'Competing Interests' statement, including any COIs declared by your co-authors. If you have no competing interests to declare, please state "The authors have declared that no competing interests exist". Otherwise please declare all competing interests beginning with twhe statement "I have read the journal's policy and the authors of this manuscript have the following competing interests:"

3. Please provide separate figure files in .tif or .eps format only and remove any figures embedded in your manuscript file. Please also ensure that all files are under our size limit of 10MB.

4. We noticed that you used "not shown" in the manuscript. We do not allow these references, as the PLOS data access policy requires that all data be either published with the manuscript or made available in a publicly accessible database. Please amend the supplementary material to include the referenced data or remove the references.

5. We have noticed that you have uploaded Supporting Information files, but you have not included a list of legends. Please add a full list of legends for your Supporting Information files after the references list.

Additional Editor Comments (if provided):

Reviewers' comments:

Reviewer's Responses to Questions

**Comments to the Author**

1. Does this manuscript meet PLOS Digital Health’s publication criteria? Is the manuscript technically sound, and do the data support the conclusions? The manuscript must describe methodologically and ethically rigorous research with conclusions that are appropriately drawn based on the data presented.

Reviewer #1: Yes

Reviewer #2: Yes

Reviewer #3: Yes

2. Has the statistical analysis been performed appropriately and rigorously?

Reviewer #1: Yes

Reviewer #2: Yes

Reviewer #3: Yes

3. Have the authors made all data underlying the findings in their manuscript fully available (please refer to the Data Availability Statement at the start of the manuscript PDF file)?

Reviewer #1: Yes

Reviewer #2: No

Reviewer #3: Yes

4. Is the manuscript presented in an intelligible fashion and written in standard English?

Reviewer #1: Yes

Reviewer #2: Yes

Reviewer #3: Yes

5. Review Comments to the Author

Reviewer #1: This study looked at an important topic and evaluated the drug dosage modifications in a chart reviews about polypharmacy. The study has been conducted in a methodologically sound way. However, given that there has only been a handful of retrospective and prospective studies on this topic.

I think the main contribution of this study is to show the limited evidence on the polypharmacy with limitations about EHRs and their modifications. It will add much to literature.

Major concerns:

1. Polypharmacy concepts in western populations needs to be defined in a age specific definitions. It would have been better, if authors might have emphasized more on the introduction part about polypharmacy and stressed why this methodology was used. I am impressed by using Bayesian inferences and adjusted ORs given the complexity of drugs dosages.

2. Please include Clinical meaningful benefits of this study findings in discussion and conclusion sections, respectively. 

3. Reconsider and please rewrite the discussion with respect to your study findings.

Thank you!

Reviewer #2: In this manuscript, the authors develop a model to characterize the complexity of polypharmacy and identify concomitant drug pairs subject to more frequent dosage adjustments. They also describe their association to outcomes, diagnoses, and laboratory values and classify co-medication pairs based on whether they have been labelled as DDIs previously and share pharmacokinetic properties. I do believe that the paper is both interesting and useful. I have no major concerns in regards to methodology, but do suggest that the authors should release their data and source code so that the readers can follow their work or check it.

Reviewer #3: Firstly, congratulations to the authors on the successful submission of their paper for publication. The manuscript is well written and demonstrates a sound methodology.

A few general comments:

1. I suggest that the authors consider reorganizing the abstract to enhance its clarity and highlight the manuscript's contribution. It would be ideal to begin with the background and hypothesis, followed by a description of the methodology, and conclude with a brief summary of the results and their potential implications.

2. I recommend reorganizing the "Results" section for improved readability. Specifically, the first paragraph under "Condensation of millions of prescriptions into drug dosage patterns" could be more appropriately placed within the "Methods" section.

3. It would be beneficial to divide the "Discussion" section into sub-sections if feasible.

4. I suggest the authors consider increasing the font size of test in figures wherever possible. 

Critical comments:

1. It is necessary for the authors to provide a justification for the use of a non-parametric test in their study. I recommend examining the variable distributions for normality such as the Kolmogorov-Smirnov (KS) test. Additionally, when presenting results in terms of p-values, I encourage the authors to report effect sizes such as Cliff's d or Cohen's d. 

2. It would be ideal to provide a rationale for assuming a normal distribution for the specified set of priors in their study. It is also important to discuss the potential implications if a normal distribution assumption does not hold.

6. PLOS authors have the option to publish the peer review history of their article (what does this mean?). If published, this will include your full peer review and any attached files.

**Do you want your identity to be public for this peer review?** For information about this choice, including consent withdrawal, please see our Privacy Policy.

Reviewer #1: Yes: Rakesh Kumar, M.D.

Reviewer #2: No

Reviewer #3: No

---

## [Editor Report · Decision Letter 1]

20 Jul 2023

Drug dosage modifications in 24 million in-patient prescriptions covering eight years:  A Danish population-wide study of polypharmacy

PDIG-D-23-00017R1

Dear Professor Brunak,

We are pleased to inform you that your manuscript 'Drug dosage modifications in 24 million in-patient prescriptions covering eight years:  A Danish population-wide study of polypharmacy' has been provisionally accepted for publication in PLOS Digital Health.

Best regards,

Benjamin P. Geisler, M.D., M.P.H., F.A.C.P., M.R.C.P. (London), F.H.M.

Academic Editor

PLOS Digital Health

Note: there are no specific peer reviewer or editor comments.